# A Qualitative Study Supporting Optimal Nutrition in Advanced Liver Disease—Unlocking the Potential for Improvement

**DOI:** 10.3390/nu16152403

**Published:** 2024-07-24

**Authors:** Shaye Ludlow, Katherine Farragher, Kelly Squires, Susan Heaney, Jessica Orman, Sarah Pullen, John Attia, Katie Wynne

**Affiliations:** 1John Hunter Hospital, Hunter New England Local Heath District, New Lambton Heights, NSW 2305, Australia; shaye.ludlow@health.nsw.gov.au (S.L.); jessica.orman@health.nsw.gov.au (J.O.); sarah.pullen@health.nsw.gov.au (S.P.); john.attia@newcastle.edu.au (J.A.); 2Hunter Medical Research Institute, Equity in Health and Wellbeing, New Lambton Heights, NSW 2305, Australia; 3School of Health Sciences, University of Newcastle, Callaghan, NSW 2308, Australia; katherine.farragher@uon.edu.au (K.F.); kelly.squires@newcastle.edu.au (K.S.); 4Department of Rural Health, University of Newcastle, Port Macquarie, NSW 2444, Australia; susan.heaney@newcastle.edu.au; 5School of Medicine and Public Health, The University of Newcastle, Callaghan, NSW 2308, Australia

**Keywords:** liver disease, malnutrition, quality of life, social support, supportive care, palliative care, screening, assessment

## Abstract

Malnutrition rates in Advanced Liver Disease (ALD) are significantly higher than those in well-compensated liver disease. In addition to its physiological impact, malnutrition is detrimental for quality of life and social, emotional, and psychological well-being. Studies within oncology and renal supportive care have identified the influence of non-physiological factors on malnutrition risk. Integrating similar factors into malnutrition screening for ALD could improve identification of at-risk patients to optimize treatment planning. This qualitative study aimed to understand the holistic factors influencing nutritional status in the ALD population. Semi-structured interviews with 21 patients, carers, and clinicians explored the experiences of malnutrition in ALD. Thematic analysis revealed five key themes: (i) appropriateness of healthcare delivery; (ii) health- and food-related factors; (iii) high symptom burden, (iv) social support impacting well-being, and (v) physical and structural supports. Current screening methods do not adequately capture all potential drivers of malnutrition in the ALD population. Adopting a more supportive approach including both physiological and non-physiological factors in ALD malnutrition screening may promote more timely and comprehensive nutritional interventions that address the complex and holistic needs of patients living with ALD.

## 1. Introduction

Malnutrition in Advanced Liver Disease (ALD) is common and contributes to high symptom burden, poorer health outcomes, increased hospital admissions, increased healthcare costs, decreased quality of life, and increased rates of mortality [1,2,3,4,5,6]. Studies have shown that 20% of people living with well-compensated liver disease suffer from malnutrition, with rates increasing to 60–85% in people in advanced stages of the disease [1,2,3,6]. Malnutrition directly impacts severe complications of liver cirrhosis, such as ascites, hepatic encephalopathy, and infections, worsening prognosis and reducing quality of life [6]. 

Early detection and diagnosis of malnutrition is essential in ALD. The European Society for Clinical Nutrition and Metabolism (ESPEN) guidelines recommend initial screening for malnutrition using a validated tool [1,2,3,6,7]. While most globally used validated screening tools seek to identify changes in dietary intake and weight, screening for malnutrition in ALD can be difficult. The fluid buildup from edema and ascites may mask weight changes, leading to inaccurate weight and body mass index calculations and thereby decreasing the accuracy of these tools and potentially underestimating malnutrition risk [6,7,8]. As a result, nutritional intervention is often delayed [6]. The Malnutrition Screening Tool (MST) is an example of a screening tool commonly used in clinical practice [3]. It is a popular tool as it is validated across multiple conditions, can be completed by any staff member, and can be easily applied in all settings [3]. However, it is less likely to be effective in this population as it only uses intake and weight as indicators of risk of malnutrition [3]. Some studies have shown that in patients with cirrhosis, the MST has poor diagnostic ability, with reduced sensitivity and lower prevalence of identified malnutrition risk when compared with other screening tools [2].

The Patient-Generated Subjective Global Assessment Short Form (PG-SGA Short Form) includes a wider range of nutritional symptoms such as nausea, taste change, and loss of appetite, with the option to include additional factors under the category of ‘other’ [9]. However, this ‘other’ information is reliant on a patient’s health literacy or understanding of the condition to recognize key details to convey for effective identification and prioritization of need [9]. Alternatively, the Royal Free Hospital-Nutrition Prioritizing Tool (RFH-NPT) is recommended by the European Society for Enteral and Parenteral Nutrition (ESPEN) guidelines for malnutrition screening in patients with ALD as it is disease-specific and considers factors beyond intake and weight [1,3]. Included in the tool are disease-related complications such as ascites and general fluid overload, therefore improving its effectiveness for malnutrition risk detection in this population [6,10].

Non-physiological determinants such as personal, religious, and cultural values and psychological factors play a role in malnutrition etiology but are not routinely captured in current tools [3,6,7,8,11,12]. While there is limited evidence investigating the non-physiological determinants of malnutrition in ALD populations, studies in other patient groups with comparable high symptom burdens, such as oncology and renal supportive care, have shown that incorporating these additional factors leads to more timely nutritional intervention for those at risk of malnutrition [11,12]. This allows for better management of symptoms, improved physical function and independence, and improved quality of life [11,12].

Given the significant impact of malnutrition on ALD outcomes and emerging evidence regarding the benefit of incorporating non-physiological determinants of malnutrition in screening for ALD, a broader approach to malnutrition screening is required. The primary aim of this research is to identify these additional non-physiological factors for use in malnutrition screening for patients with ALD that could inform a patient-reported screening tool. Consumer engagement explored the lived experience of nutrition concerns in this patient group through qualitative methods. Such consumer involvement has been shown to increase the quality, impact, and reach of clinical research [13].

## 2. Materials and Methods

This qualitative study is reported in accordance with the ‘Standards for Reporting Qualitative Research (SRQR): A Synthesis of Recommendations’ [14]. Ethics approval was provided by Hunter New England Ethics Committee reference (2022/ETH02426) on 21 December 2022. Registration was granted by the Australian New Zealand Clinical Trials Registry (ANZCTR), ACTRN12624000472572.

### 2.1. Setting

This study was conducted within complex and supportive care liver clinics across one metropolitan and one regional Australian public health site. 

### 2.2. Research Steering Committee

The research team comprised a ten-member steering committee consisting of researchers, doctors, nurses, and dieticians with experience in ALD. An early-career research dietician skilled in working with ALD patients and supportive care convened the research steering committee. The research dietician worked collaboratively under the guidance of the broader research steering committee, seeking feedback and direction on every aspect of the study design, implementation, and evaluation.

### 2.3. Consumer Input

For this study, it was vital to engage with a range of consumers to improve the researchers’ understanding of the lived experience of malnutrition in ALD and to ensure the richness of the data. Consumers consisted of (i) patients and (ii) carers.

Patients were eligible if they were 18 years of age or older, attended one of the liver clinics in the research setting, were able to provide informed consent, and had Child–Pugh B or C liver disease. As a range of lived experience was sought, current malnutrition was not an inclusion criterion for the study; participants were invited if they had Child–Pugh B or C status as this group would be more likely to have a current or future risk of undernutrition and receive dietetic input. Carers of any patient meeting the eligibility criteria were invited to participate, with their participation dependent on receiving informed consent from the patient.

### 2.4. Clinician Input

Dieticians, nurses, social workers, and doctors who worked across either research site were eligible to participate. These health workers needed to be currently employed at one of the liver clinics in the setting, caring for patients who met the eligibility criteria and/or have experience working with the ALD population.

### 2.5. Recruitment

Recruitment occurred between January and March 2023. Eligible patients and carers were identified from those who attended a clinic within the research setting. In consultation with site leads for the Department of Nutrition and Dietetics, Gastroenterology and Supportive Care, the research dietician identified key staff members with experience in ALD who were purposefully sampled and invited to participate. Due to a limited number of dieticians available at the regional site, dieticians from an outlying health campus with expertise in ALD were nominated by department leads to provide a broader range of input and regional perspective. Clinicians who provided consent to participate were provided an information sheet prior to contact with the research dietician.

### 2.6. Data Collection

Semi-structured interviews were conducted with all consumers between January and March 2023. Interviews were facilitated by the research dietician (blinded for peer review) either in person, via phone, or virtually using Microsoft Teams. Interview guides were developed to promote consistency in the data collected (see Table 1 and Table 2) based on the literature and the research team’s knowledge and clinical experience. Thirteen clinicians were interviewed either individually or in small groups of two or three. All patients and carers were interviewed individually. 

Adroit Transcription was used to transcribe a total of eleven interviews with thirteen participants, including two focus groups [15]. A glossary of terms was provided to transcribers to give context to the interviews for more accurate transcription (see Appendix A). The remaining nine participant interviews were transcribed manually by the research team due to poor recording quality. Each transcript was then checked by two members of the research team (blinded for peer review) for accuracy. 

### 2.7. Data Analysis

Transcripts were de-identified and analyzed thematically using the Braun and Clarke method [16]. The initial coding was completed by two members of the research team (blinded for peer review) who spent time familiarizing themselves with the transcribed interviews by completing iterative readthroughs of each interview to gain maximum insight [16]. Codes were generated for as many topics as possible, ensuring that the code was not just a phrase but rather applied to a contextual segment of the interview. Two members of the research steering committee (the dietetic honors student and one senior research dietician) generated themes from the existing codes and sorted these into categories that best encompassed their meaning [16]. Themes were further refined during two consultations with the broader research steering committee for review, revisiting and reorganizing codes and subthemes until a consensus was obtained [16]. The sample size was deemed large enough to provide adequate information power due to the narrow focus of the study, specific sample population, and clinical expertise of the research team [17]. Supporting exemplars were then extracted from the text, aiding in the refining of the names of the themes, ensuring that they were conceptually parallel [16]. NVivo 12 software was used to assist in the organization and visualization of the data [18]. 

## 3. Results

There was a total of twenty-two participants included in this study: eleven clinicians, nine participants, and two carers. Five clinicians and two patients came from the regional sites. Six clinicians, seven patients, and two carers came from the metropolitan site (see Figure 1). Patient participants had a diagnosis of hepatitis B, hepatitis C, and/or alcoholic liver disease and had received dietician input into their care planning during standard care liver clinics in the six months prior to being recruited to the study. Two of these patient participants had been diagnosed with hepatocellular carcinoma, with one having received immunotherapy. There was a 100% participation rate with all patients, carers, and clinicians approached for the study agreeing to be interviewed. Interviews lasted between five and thirty minutes.

When exploring the factors influencing the nutritional status of the ALD population, five main themes were identified: (i) appropriateness of healthcare delivery; (ii) health- and food-related factors; (iii) high symptom burden, (iv) social support impacting well-being, and (v) physical and structural supports (see Table 3 for themes, sub-themes, and supporting quotes).

### 3.1. Appropriateness of Healthcare Delivery

Timely referrals and capturing and addressing what matters most to patients and their carers were highlighted by carers and clinicians as important factors in this study. Patients recognized the benefit of being able to “ring up and ask a question” and carers identified “having the right group around you, especially a group that sort of specializes in that stuff…when no-one else really took very serious, so um, life-changing, for both of us”. Clinicians acknowledged that it was difficult to assess malnutrition risk with weight changes alone due to an “increase in ascites [and] decrease in muscle mass”. This highlighted the need for a screening tool that could screen for more symptoms specific to the ALD population to allow for more appropriate referrals to dieticians. Clinicians identified that the current method of screening for malnutrition was “just a bit of a hit and miss” without “a real clear process of who gets referred to the dietician” and “not capturing the issues that the patient has”.

### 3.2. Health- and Food-Related Factors

One patient identified that having a “common understanding of nutrition” would help increase their intake as they would then be able to understand “the goodness [they’re] getting from the food to make it worthwhile eating”. For some patients, it was recognized by clinicians that “they’ve not had good, potentially, dietary or nutrition inputs until now and so those [negative] habits are formed”, as well as “a lot of people we do know still have a reliance on alcohol as well” which can then make it “hard to break that barrier as well, sometimes.” Clinicians noted that this impeded some patients’ comprehension of the provided education regarding how food and nutrition may impact ALD. 

It was identified by a clinician that there are “multi-factorial” components that can impact the nutrition of people with ALD. Clinicians reported the patients’ “cooking skills and food preparation skills are, on average, lower in this population” and could adversely affect their nutritional status. Additionally, it was recognized that aspects such as “issues with dentition” could impact their nutritional status as it “changes what people can eat and what they can tolerate, that links in often with financial things, not being able to get their teeth fixed, or access the services that they need.” This comment reflects difficulties patients have accessing public dental health services in Australia due to the associated costs, waiting times, limited transport options, and fewer dental practitioners in rural areas [19].

### 3.3. High Symptom Burden 

It was identified by patients, carers, and clinicians that the common physiological symptoms associated with ALD, such as ascites, nausea, vomiting, poor appetite, and loss of taste, could affect a patient’s ability to maintain good nutrition. This was best described by one patient as the “absolute sickness” that would get in the way of their nutrition. 

While for some patients the symptoms were unexpected without an apparent specific cause, for others, symptom burden was recognized as being complex and often interrelated, causing a cascade effect on intake: “I was eating perfectly, and then one day I just decided I didn’t feel like eating anymore” and “Quite often when I eat, I throw up. Just out of the blue”. As a result of these symptoms, there was a loss of interest in food, making eating more challenging and akin to an obligatory task. The motivation to eat could be quite transient and subject to frequent changes. 

For some patients, despite best intentions, the experiences of symptoms impacted “that motivation to cook” and to “actually prepare the meals”, as highlighted by one patient after frequent bouts of diarrhea impacted their ability to proceed with planned meals. Additionally, one clinician reported that “80 per cent of them [patients] are probably depressed,” suggesting that low mood would further impact motivation and ability to look after one’s nutrition. As a result of experiencing symptoms, patients’ reduced energy levels were recognized as impacting their ability to prepare meals; this began even before the meal preparation stages, in being able to source food from the shops and get it home. “Physically, I couldn’t walk. I literally couldn’t lift my three-month-old son up at the time, I barely could move off the lounge. I just slept all day”.

### 3.4. Social Support Affecting Well-Being

Patients, carers, and clinicians identified the significance of social support for people living with ALD. Patients with ALD reported a loss of independence and the need for more support from family, friends, and carers, as exemplified by one patient’s statement: “I was a very independent person, I did everything on my own, I never asked for help, I was very stubborn… Whereas now, I’m like, ‘yes please, can you do this for me’. So that’s kind of what’s changed me a lot I suppose”.

However, patients also recognized that their waning interest in food sometimes led them to decline social invitations for fear of bringing a negative atmosphere, resulting in missed opportunities for a supportive social dining experience. One patient responded that “I didn’t attend my grand-daughter’s um, engagement party on Saturday, just because you know, I hadn’t been eating and also too, because I am getting tired because also you don’t eat so I’m “I’ve got to sit down, I’ve got to sit down”, you know I’d be forever feeling I was being that wet towel, always hanging around, so I just said I wasn’t going to go and so I missed out on that.” 

Clinicians reported they noticed “a big difference in the patients who have a family, or you know, have a partner and patients who are single and are on their own”. When patients had a “lack of social support…especially if they are living on their own and they’re feeling so poorly”, this would impact a patient’s ability to maintain good nutrition due to difficulty shopping for food and preparing meals. 

### 3.5. Physical and Structural Support

Financial concerns were raised by some patient participants, with one patient responding that “I’m on the DSP [Disability Support Pension] … I don’t work at the moment, so I don’t have that money to spend”. It was identified by clinicians that a number of barriers could affect a patient’s ability to maintain good nutrition; these were recognized as patients of “lower socioeconomic [status] as well, so financial barriers, transport barriers, things like that”. 

In the clinicians’ experiences, finance had a big impact on “being able to afford to eat the amount of protein or the amount of food”. This was reinforced by patients when they were required to purchase specialty items to manage their condition or when patients were required to access nutritional support to ensure they could meet their nutritional requirements. In some instances, clinicians stated that patients would respond “no” when asked if they could afford to purchase nutritional supplement drinks. As a result, the dieticians were altering recommendations to meet the financial needs of the patients.

## 4. Discussion

This novel study aimed to understand the non-traditional factors influencing the nutritional intake of the ALD population. Our findings highlight several barriers and non-physiological factors not included in standard malnutrition screening tools that influence the nutritional status of patients in this population. 

Results from this study reinforce the impact of a high symptom burden in ALD on a patient’s nutritional status. Symptoms such as ascites, nausea, vomiting, poor appetite, loss of taste, and low energy acted as barriers to maintaining adequate nutrition. Our study also highlighted the multifactorial nature of these symptoms in patients, which were often linked; for example, accumulation of ascites may cause nausea, resulting in changes to appetite and intake. This is consistent with the literature, which indicates that up to 50% of people with ALD will experience ascites and its consequent impact on nutrition, leading to heightened malnutrition rates of up to 85% in this population [1,20]. Despite these symptoms being commonly reported in our study population and consistently seen in the literature, only two of the seven validated malnutrition screening tools, the RFH-NPT and the PG-SGA Short Form, explore disease-related complications, such as ascites or general fluid overload [6,9,20,21]. In addition, the PG-SGA Short Form includes nutritional symptoms such as nausea, taste change, and low appetite, but doesn’t capture fluid buildup from ascites or edema [9].

Evidence in the renal and oncology literature has identified that targeting non-physiological impacts can lead to better-managed symptoms and a greater quality of life [11,12]. In Australia, the PG-SGA Short Form is often preferred in renal and oncology settings. However, our study identified that patients in the ALD population have other complex needs not routinely picked up by current screening tools and experience non-physiological factors that impact their nutritional intake, such as financial status, level of social support, and food literacy levels. 

It was highlighted by clinicians that the current screening tools used in this setting did not identify all factors that influenced the nutritional status of patients with ALD, which may impact timely and appropriate referrals for those at risk of malnutrition. As reported in the existing literature, the MST, commonly used in clinical settings, is found to have poor accuracy compared to tools such as the RFH-NPT [3,6]. Therefore, earlier detection and intervention for malnutrition, as recommended by the ESPEN guidelines, is essential to improve the quality of life and reduce the mortality rate of patients with ALD [1,6]. This study draws our attention to the importance of considering a patient-centered approach to improve outcomes and support optimal nutrition to meet the needs of patients with ALD.

To the authors’ knowledge, this is the first exploration of malnutrition screening in ALD encompassing the important perspectives of patients, caregivers, and clinicians utilizing a collaborative and evidence-based approach with formal consumer engagement. As a result of their experiences, this study highlights the ineffectiveness of current screening methods and the need for more comprehensive malnutrition screening. Leveraging the consumer and clinician insights obtained in this study can inform clinical practice towards malnutrition screening that is consumer-focused and considers the unique lived experiences of this patient population. Designing effective patient-reported outcome measures could complement anthropometric or other measures to detect sarcopenia and biochemical methods of assessing nutritional status, for example, serum albumin level, total cholesterol level, and peripheral lymphocyte counts or the ‘Controlling Nutritional Status’ (CONUT) score [22]. It is important to acknowledge there was an unequal distribution of participants between the metropolitan and the regional sites. While it is not anticipated that this would have altered the themes identified in the study, it would be optimal for future research to further explore the experiences of regional and rural people with ALD in more depth due to the known health inequalities experienced by people in these regions [23]. This study did not include patient participants with Child–Pugh A status, and although it is anticipated that this would not have changed our themes, this could be another area of future study. Consideration of the health literacy of patients with ALD and the factors that may differentially influence meal frequency or meal quality may also benefit from future research. Additionally, while this study explored the experiences of patients, carers, and clinicians, there was an uneven distribution in the final numbers. As a result, there might be a lack of representation from specific groups, potentially impacting the comprehensiveness of the study [24]. Respondent validation of the findings could further support the data [24]. 

Future research in this area should focus on a supportive and holistic approach to nutritional management that considers economic, environmental, and psychosocial factors. The authors plan to develop a novel screening tool for ALD that will be evaluated in comparison to a comprehensive subjective global assessment. Furthermore, interdisciplinary care planning should be engaged to harness expertise in the management of both physiological and non-physiological determinants of malnutrition, which may improve not only the nutritional status of patients but also their quality of life.

## 5. Conclusions

Malnutrition rates in patients with ALD are higher when compared to patients with well-compensated liver disease. This study aimed to understand the non-traditional factors perceived to influence the nutritional status of patients with ALD which are not currently included in malnutrition screening tools. A wide range of physiological and non-physiological factors that can affect the nutrition status of patients with ALD were identified by the diverse group of stakeholders; these warrant consideration in the development and implementation of malnutrition screening tools for ALD. To our knowledge, no current published screening tools consider both the physiological and non-physiological factors of malnutrition in the ALD population. Therefore, we recommend the development of a novel screening tool that incorporates these aspects to screen for malnutrition in this population more accurately.

## Figures and Tables

**Figure 1 nutrients-16-02403-f001:**
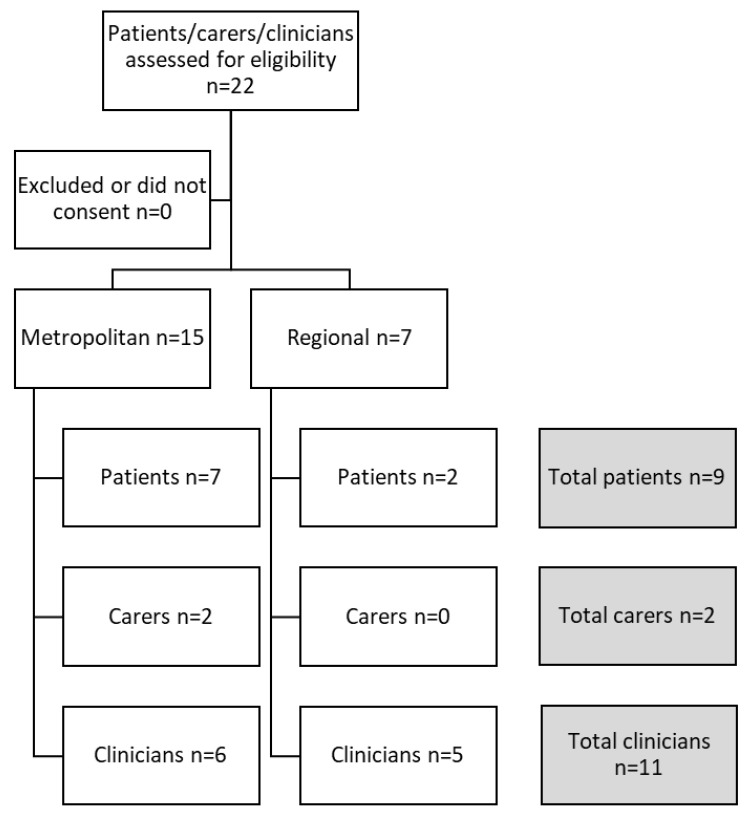
Flow chart for the recruitment of participants.

**Table 1 nutrients-16-02403-t001:** Interview Guide for Clinicians.

Welcome and Introductions.
Experiences of nutrition care in complex liver clinics.
What is your experience of using nutrition screening and assessment tools in standard practice? Can you explain the referral processes to the dietician currently used in standard care? Can you explain how these processes are effective or ineffective in identifying those patients requiring nutrition support? From your point of view, what are some of the factors influencing an ALD patient’s nutrition? How do current nutrition screening tools account for these factors? Can you explain how non-physiological factors (for example, food security, culture, dentition, psychosocial and emotional well-being) influence nutrition for patients living with ALD?
Positive/negative comments.
Can you explain the positive components of the nutrition screening tools currently used in standard practice? What works well? Can you explain the negative components of the nutrition screening tools currently used in standard practice? What doesn’t work well?
Feedback and suggestions for screening tool development.
What are your suggestions on how the screening of nutrition in ALD patients could be improved? How could the non-physiological determinants of nutrition be incorporated into a new nutrition screening tool for ALD patients? What are the main physiological and non-physiological components or considerations you think are important to include in the novel screening tool?
General suggestions.
Do you have any general recommendations? Do you have any other comments?

**Table 2 nutrients-16-02403-t002:** Interview Guide for Patients and Carers.

Welcome and Introductions.
We know that nutrition can affect more than just your physical health. It can also be important to people’s connection with their family, quality of life, culture and social and emotional well-being. We’d like to ask you a few questions about nutrition, which will help us improve the way we plan and provide nutritional care
Experiences of care.
Can you tell me how nutrition affects your daily life? Other than your physical health, can you explain how nutrition impacts other areas of your well-being?
Positive/negative comments.
What are the things that make it hard for you to have good nutrition? What are the things that might help you have good nutrition?
Feedback.
What things do you think are important for the healthcare team to consider when asking about your nutrition and how it affects your daily life?
General suggestions
Do you have any general suggestions or comments?

**Table 3 nutrients-16-02403-t003:** Themes, sub-themes, and supporting quotes.

Theme	Sub-Theme	Demonstrative Quotes—Patient or Carer	Demonstrative Quotes—Clinician
Appropriateness of Healthcare Delivery	Accurate Screening/Correct Referrals	“Yeah well didn’t have the right people around her. I suppose it was month in, month off, every six weeks she’d end up in [de-identified] Hospital, um, she’d make little improvements, but there was no eating involved. Ah, she made little improvements then they’d send her home. Um, and you’d just wait five or six days and we’re on the downhill slide again. Look, the only thing that changed (de-identified) life was meeting that lady in the waiting room that day and getting (de-identified) in the liver clinic, or my daughter would be dead.” (carer)	“They might have already been in the hospital for say, five days, before we even know about them and I think that really is a reflection of the MST, not capturing the issues that the patient has.” (clinician) “If there was an effective screening tool, that would obviously create more referrals to a dietician, or more appropriate referrals, potentially. What I know across the hospital, is a malnutrition screening tool is, not sensitive enough to pick up the people who really need it or often results in multiple referrals for people who don’t really need to see a dietician at all, so I think it really needs to be developed for the population that we are looking at, at the time, or using something more in depth than just a MST” (clinician) “Very difficult with weight changes-increase in ascites [and] decrease in muscle mass.” (clinician) “it’s just a bit of a hit and miss…it doesn’t seem to be a real clear process… [of] who gets referred to the dietician”. (clinician)
	Supportive Care Team	“If I needed to ring up and ask a question about if I could eat something or not, eat something which is great, like you guys jumped on. If you guys didn’t know. I know you personally look up for me whether I should have it or how much I would like to have that.” (patient) “I was (de-identified) primary carer and the kids’ primary carer, I had somebody…you blokes were ringing me up a couple of times per week, just to make sure how things were going and that. Do you know how much benefit that did for me? Do you know how much benefit that gave (de-identified)? Knowing somebody gave a s**t…but you blokes are liver specialists and dieticians and doctors and…mental health person, you know she was beautiful too. Having the right group around you, especially a group that sort of specialises in that stuff, if you like, when no-one else really took it very serious, so um, life changing, for both of us.” (carer)	“entering a more…transient approach…being in the same room, or sharing notes, getting a bigger picture of what’s happening in the world and bouncing off each other always seems to work really well.” (clinician)
Health- and Food-Related Factors	Dentition	“had to have a root canal the other week. So they’ve been pretty bad and there’s more to be done there…I don’t have the money, but I don’t want to go around with no teeth either.” (patient) “they said if you don’t…get that tooth fixed, you won’t be able to eat on that side of your mouth.” (patient) “[food] gets stuck all over my dentures…I gotta keep going to the bathroom and cleaning my teeth out.” (patient)	“We’ve had patients with dental issues that’s impacted their nutrition” (clinician) “issues with dentition” (clinician) “changes to dentition of course changes what people can eat and what they can tolerate, that links in often with financial things, not being able to get their teeth fixed, or access the services that they need” (clinician)
	Food Literacy	“You know, last night I had a bit of cheese and flavoured milk and all that and that’s always good, you know I’ll drink flavoured milk rather than drink an apple juice, or something like that. yeah, yeah, so good education and meal suggestions [would help improve nutrition]” (patient) “And the goodness [they’re] getting from the food to make it worthwhile eating, hmm. Like common understanding of nutrition.” (patient)	“I’d say, like cooking skills and food preparation skills are, on average, lower in this population”. (clinician) “the health literacy of a lot of the patients that come to clinic and I guess… it has to be balanced… a lot of patients would struggle or lose interest or not understand.” (clinician) “they’ve not had good, potentially, dietary or nutrition inputs until now and so those [negative] habits are formed.”(clinician) “I guess general knowledge about health because they probably, I know it’s very stereo-typical but, um, they’ve not made positive, healthy, lifestyle choices, so it’s hard to break that barrier as well, sometimes.” (clinician)
	Alcohol Consumption	“I had wine and there was two and then there was three and then there was four long drinks. I’d drink right up until news was over at 7 o’clock, cook some dinner and have one bite and put it in the fridge, saying I’d have it tomorrow and it’d end up in the bin, so the alcohol was probably sustaining my hunger perhaps, I don’t know, but it put me in the wrong frame of mind to get some tucker in to me and feeling healthy.” (patient)	“There can often still be alcohol consumption, which can affect their nutrition.” (clinician) “a lot of people we do know still have a reliance on alcohol as well, which can impact their nutritional status, so I guess it’s multi-factorial.” (clinician)
High Symptom Burden	Sickness	“Oh, um, well at the moment, ‘cause being sick and all that, sort of affects [diet] probably different ways than what you’d probably expect.” (patient) “it’s the absolute sickness.” (patient) “She spent more time with her head in a bucket, than she did with her head on a pillow” (carer)	“stage of disease [influences nutrition].” (clinician) “the disease state is probably a big thing, in progression.” (clinician)
	Unexpected Symptoms	“I was eating perfectly and then one day I just decided I didn’t feel like eating anymore and yeah food just didn’t interest me at all like nothing and it took a very long time to get that interest that I wanted to eat back that’s for sure” (patient) “I get full pretty quick.” (patient)	“physiological things, ascites, nausea, vomiting, poor appetite, loss of taste,” (clinician)
	Cascade	“And quite often when I eat I throw up. Just out of the blue. But with some nausea in the beginning or just randomly after eating… I ate and it makes it worse, because you’re feeling so uncomfortable.” (patient)	“a lot of people with liver disease, their ascites can cause nausea and… shortness of breath and things as their ascites can build up…and impact their diet in that way and changes to their bowels.” (clinician)
	Loss of Interest in Food	“Yeah, that most people find it a really pleasurable thing whereas it’s a bit of a chore for me [eating].” (patient) “I do get a funny taste in my mouth…it’s like a filter’s been put over my tongue and my tastebuds. If I want to try and enjoy something and that filter’s over there and I can’t taste it, I think well what’s the point?” (patient) “I went to the butcher today and I bought all this stuff. I thought, well I’m really good at buying food, sometimes I’m good at preparing it, I’m just not good at eating it” (patient) “No appetite, like I literally have to make myself eat” (patient) “The ascites, I think, had a lot to do with it and having the NG tube made it harder to eat for me.” (patient) “He’s gone off meat and everything like that, so it’s hard to get him interested in food, to get him to eat.” (carer)	“I think, a lot of the time I think there is loss of appetite.” (clinician) “The ascites is when they’re really quite bloated, they find it difficult to eat.” (clinician)
	Loss of Motivation	“It’s just…it’s just the energy level, I just can’t be bothered making…I bought that Light and Easy, but I’ve gone off that, so…I really don’t know what I’m going to do for…I just can’t be bothered, type of thing.” (patient) “Sometimes it’s day three of diarrhea and I feel rotten, so I can plan that sort of stuff but as far as a meal goes, food goes, I, plan it but…I won’t go ahead with it.” (patient) “When I’m in a care situation and food’s put in front of you…it’s easier to do that. I need to get the motivation to cook for myself and enjoy that again.” (patient)	“people may not have always had healthy eating habits and but it’s also actually identifying do they actually see that as a problem, do they want help with that”. (clinician) “that motivation to cook and to…actually prepare the meals can be limited as well”
	Confusion	“you’re just too confused to even think about food.” (patient)	“When they [are] confused, they just don’t wanna eat. They get to the point where, yeah, they just don’t wanna eat.” (clinician)
	Loss of Energy	“Physically, I couldn’t walk. I literally couldn’t lift my three-month-old son up at the time, I barely could move off the lounge. I just slept all day. I had no energy whatsoever… I couldn’t go to the shops, I couldn’t carry any bags I couldn’t take the kids to the park or out to the beach or anywhere like that because I physically couldn’t walk more than 5 meters” (patient) “My body’s not absorbing protein so that makes me really tired and then you get tired and it’s just like that wicked circle again, you get tired, you can’t be bothered you just don’t do and I’m just trying to step over that at the moment” (patient) “she didn’t have the strength, the energy, she didn’t have um, the desire.” (carer)	“One of the key symptoms that you see from the liver clinic which probably 90 per cent of people experience is lethargy and so preparing something that’s nutritious and that doesn’t come out of a packet somewhere or from a take-away store is…takes energy and that can steal something from their ability” (clinician) “Their, energy levels, and you know, um, their desire to eat and prepare food.” (clinician)
	Depression	“When I’m depressed I don’t eat.” (patient) “Not having the nutrition that I needed to have, it depressed me a lot, it gave me depression I couldn’t even take my daughter to school, I couldn’t be there for her year 6 formal, yeah so my family missed out on a lot with their mum I guess as well like they suffered a lot for me being sick.” (patient) “If you’re not eating and you’re not getting that nutrition, you’re not feeling like wanting to do anything. And then that affects your mental health, like it’s a you know a horrid cycle.” (patient)	“I think sometimes some of these people, like, you’d say 80 per cent of them are probably depressed…and that’s got to have an impact on, on motivation.” (clinician)
	Varices		“Often may have swallowing difficulties or dysphagia related to varices” (clinician)
Social Support Impacting Well-being	Need for Support	“I do online shopping and I make it for a day and a time that I know that someone else is going to be here because they drop it to the door, which is great, but then I’ve got to get it from the door to the kitchen and I can’t make it.” (patient)	“I think it’s important to know their level of support, at home, like carers…like meals on wheels, things like that, are they using any services, what services are there that can help.” (clinician) “Sometimes don’t have good support networks so not only do they often forget or not hear the messages being given clearly during clinic they don’t have that network to help keep them on the right path once at home and assist when needed/having a low health day.” (clinician)
	Loss of Independence/Challenges with Independence	“I was a very independent person, I did everything on my own, I never asked for help, I was very stubborn… Whereas now, I’m like, ‘yes please, can you do this for me’. So that’s kind of what’s changed me a lot I suppose.” (patient) “My dad took care of me and my kids and did a lot of the grunt work for me cause I couldn’t physically do it but he helped a lot and he helped a lot to get me to eat, he’d get me food that I thought I’d like to eat and not forcing me but re reassuring me that I had to eat pretty much that was the only way I was going to get better.” (patient) “I ended up moving me and my kids in with my dad to support me for a few months that I couldn’t eat and couldn’t function properly and so much. So you take my kids to school and run me to the doctor appointments and things like that because I couldn’t drive because I was so malnutrition and I was so skinny.” (patient) “I was with the NG tube, I pretty much didn’t eat at all. He tried to get me to have family meals with them guys, but I’d have a mouthful and that’d be enough. And gradually got to the point where I’d have lunch with them, and then I would have a family dinner. But I’ll be little and then gradually got more and more. He’d buy me vanilla slices and things like that just to try and fatten me up a little bit and it worked and it definitely worked.” (patient) “If I didn’t have his support too. I don’t think I would have the energy or the will to even want to cook anything to eat or go to the shops to get anything that I’d like, which where he would do them things for me, which helped a lot.” (patient) “I try to always have a decent tea, you know, even if I’ve eaten nothing or, you know, rubbish, throughout the day, I will try to have like a normal tea, where we sit down, at the table, like a family, ah, if I’m not, ah, eating, I don’t obviously, go to the table. You know, that takes away from that family sort of time.” (patient) “My daughter, she cooks meals every day and she will let me go maybe one or two days and then that’s…she’s like ‘nah, that’s it, you haven’t had this for ages’ and I’m like ‘uh, but I don’t feel like it’ and she’ll just dish me up something. And I will find, probably seven out of ten times, that I do eat it.” (patient)	“somehow feeling responsible for what’s happened to them and um, not worthy of some of the help that some people can easily access, is easily a barrier” (clinician) “lack of social support…especially if they are living on their own and they’re feeling so poorly.” (clinician)
	Social Isolation	“I didn’t attend my grand-daughter’s um, engagement party on Saturday, just because you know, I hadn’t been eating and also too, because I am getting tired because also you don’t eat so I’m ‘I’ve got to sit down, I’ve got to sit down’, you know I’d be forever feeling I was being that wet towel, always hanging around, so I just said I wasn’t going to go and so I missed out on that.” (patient)	“I mean there’s a big difference in the patients who have a family or you know, have a partner and ah, patients who are single, and are on their own, yeah because I think you know, a lot of the older, single men, really struggle.” (clinician)
Physical and Structural Supports	Socioeconomic Status	“my dad helped me financially.” (patient)	“Being able to afford to eat the amount of protein or the amount of food, you know, or get all of those medical appointments in, travel back and forth to [regional hospital]…it’s a huge impact, out here especially. (clinician) “Patients can’t afford to buy the appropriate foods, at times.” (clinician) “lower socioeconomic [status] as well, so financial barriers, transport barriers, things like that.” (clinician) “I think that food security is a bigger issue than we credit in a lot of different disease groups, that for a lot of people with um, liver disease, that could be an issue if they are no longer working, and they don’t have a huge social network that can support them.” (clinician)
	Affording Nutrition Treatment Options	“I’m on the DSP [Disability Support Pension] … I don’t work at the moment, so I don’t have that money to spend [on supplement drinks]” (patient) “The dieticians that say “have those pro-biotic drinks” and this, that and the other, and they’re not cheap, you know, like, I’ve got to provide for a family so I try and get stuff that we can all eat and so, yep, the finances would…would play a part in it. You know I buy different things for the family that I wouldn’t have just for myself.” (patient)	“They might be on a supplement here in hospital, but they simply can’t afford that when they go home so we’re often recommending things like up and go energisers, stuff like that, that they can easily pick up at the supermarket that might be a bit cheaper, so financial…those factors are, huge.” (clinician) “Socioeconomics is a big issue, you know, people being able to afford supplements.” (clinician)

## Data Availability

The original contributions presented in the study are included in the article/Appendix A; further inquiries can be directed to the corresponding author(s).

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
