# Peer review of "A Qualitative Study Supporting Optimal Nutrition in Advanced Liver Disease—Unlocking the Potential for Improvement"

_nutrients, 2024, doi:10.3390/nu16152403_

Round 1

Reviewer 1 Report

Comments and Suggestions for Authors

Authors reported the usefulness of a questionnaire to assess malnutrition in patients with advanced liver diseases. Patient reported outcome is an attractive method to evaluate patient's status including QOLs. This report is interesting and valuable. But the presentation was unsatisfactory.

1. Liver etiology is largely related to the results. Author should clarify the etiology of liver diseases and the treatments.  

2. Nutritional status should be assessed by conventional methods such as albumin, lymphocyte counts or CONUT score.

3. Sarcopenia should be also evaluated in present study.  

Comments on the Quality of English Language

English was well written. Minor spelling check is preferrable.

Author Response

Review 1

Authors reported the usefulness of a questionnaire to assess malnutrition in patients with advanced liver diseases. Patient reported outcome is an attractive method to evaluate patient's status including QOLs. This report is interesting and valuable. But the presentation was unsatisfactory.

We are grateful to the reviewer for their interest in the manuscript and their thoughtful comments. Please see our responses and details of the amendments made to the manuscript below:

  1. Liver etiology is largely related to the results. Author should clarify the etiology of liver diseases and the treatments.  

This qualitative research includes semi-structured interviews exploring consumer views on the factors that could lead to malnutrition. These consumers (n=9) had Child Pugh B or C liver disease and attended complex and supportive care liver clinics in a metropolitan and regional public health service. The reviewer points out that the aetiology and/or treatment may influence the participants responses or views around nutrition, therefore, to allow the reader to assess the generalisability of the results, we have added ‘Patient participants had a diagnosis of hepatitis B, hepatitis C, and/or alcoholic liver disease …and had received dietitian input….’ And ‘Two of these patient participants had been diagnosed with hepatocellular carcinoma, with one having received immunotherapy [lines 172-176].

  1. Nutritional status should be assessed by conventional methods such as albumin, lymphocyte counts or CONUT score.

Thank you for the comment. As you note, the aim of this study is to explore physiological and non-physiological factors that could support the development of patient reported outcome measures, which we have clarified in the introduction [lines 81-82].  We have noted in the discussion that ‘Designing effective patient-reported outcome measures could complement anthropometric or other measures to detect sarcopenia and biochemical methods of assessing nutritional status, for example serum albumin level, total cholesterol level, and peripheral lymphocyte counts or the ‘Controlling Nutritional Status’ (CONUT) score [22].’, adding the paper confirming the validity of this method doi:10.1590/S0212-16112012000200033. [lines 323-327].

  1. Sarcopenia should be also evaluated in present study.  

Thank you for emphasising that the recognition of sarcopenia in people living with advanced liver disease is crucial for prognostication, management and wellbeing. Although, measurement of sarcopenia was not a proposed component of a patient-reported outcome measure and therefore was not an element of the semi-structured interviews, we have noted in the discussion that ‘Designing effective patient-reported outcome measures could complement anthropometric or other measures to detect sarcopenia and biochemical methods of assessing nutritional status, for example serum albumin level, total cholesterol level, and peripheral lymphocyte counts or the ‘Controlling Nutritional Status’ (CONUT) score [22].’, adding the paper confirming the validity of this method doi:10.1590/S0212-16112012000200033. [lines 323-327].

Reviewer 2 Report

Comments and Suggestions for Authors

The manuscript of Ludlow et al. deals with the assessment of malnutrition rates in patients of advanced liver disease (ALD), since malnutrition is detrimental for quality of life and social, emotional and psychological wellbeing. Authors integrate non-physiological factors having influence on malnutrition risk. This qualitative study aimed to understand the holistic factors influencing nutritional status in the ALD population. Semi-structured interviews covered five key themes: (i) appropriateness of healthcare delivery; (ii) health and food related factors; (iii) high symptom burden, (iv) social support impacting wellbeing and (v) physical and structural supports. Authors concluded that inclusion of both physiological and non-physiological factors in ALD malnutrition screening may promote more timely and comprehensive nutritional interventions. This work appeared to be well performed, showing interesting data. However, I have some minor concerns:

1)    Assessment of malnutrition rate was performed in ALD with patients classified as Child-Pugh B or C liver disease; There were also tested nutrition rates in patients with Child-Pugh A? Indeed, malnutrition contributes to the advance of this disease?     

2)    Besides the data analysis, why a statistical analysis was not applied? How was calculated the “n” of patients necessary to reinforce the conclusions?

3)    The concept of malnutrition relies in the amount of ingested food, its quality or the frequency of meals?

4) It seems that the main non-physiological factors relay in a depressing behavior and in the economical status. How these factors can be analyzed in depth?

Author Response

Review 2

The manuscript of Ludlow et al. deals with the assessment of malnutrition rates in patients of advanced liver disease (ALD), since malnutrition is detrimental for quality of life and social, emotional and psychological wellbeing. Authors integrate non-physiological factors having influence on malnutrition risk. This qualitative study aimed to understand the holistic factors influencing nutritional status in the ALD population. Semi-structured interviews covered five key themes: (i) appropriateness of healthcare delivery; (ii) health and food related factors; (iii) high symptom burden, (iv) social support impacting wellbeing and (v) physical and structural supports. Authors concluded that inclusion of both physiological and non-physiological factors in ALD malnutrition screening may promote more timely and comprehensive nutritional interventions. This work appeared to be well performed, showing interesting data. However, I have some minor concerns:

Many thanks for your positive comments regarding the aim of our study. Please see our responses and details of the amendments made to the manuscript below:

1)    Assessment of malnutrition rate was performed in ALD with patients classified as Child-Pugh B or C liver disease; There were also tested nutrition rates in patients with Child-Pugh A? Indeed, malnutrition contributes to the advance of this disease?     

This qualitative research includes semi-structured interviews exploring consumer views on the factors that could lead to malnutrition to provide lived experience. These consumers (n=9) had Child Pugh B or C liver disease and attended complex and supportive care liver clinics in a metropolitan and regional public health service. To clarify our consumer input, we have added ‘As a range of lived experience was sought, current malnutrition was not an inclusion criterion for the study; participants were invited if they had Child-Pugh B or C status as this group would be more likely to have a current or future risk of undernutrition and receive dietetic input.’ [lines 107-110].

We agree that timely nutritional intervention in people with Child-Pugh A status remains important, and therefore their views as consumers may be useful in a broader study. We have noted this as a potential limitation: ‘This study did not include patient participants with Child-Pugh A status, and although it is anticipated that this would not have changed our themes, this could be another area of future study’ [lines 332-334].

2)    Besides the data analysis, why a statistical analysis was not applied? How was calculated the “n” of patients necessary to reinforce the conclusions?

This was a qualitative study with transcripts of the semi-structured interviews and exploration using the Braun and Clarke method for thematic analysis. Sample size in qualitative studies is not ascertained using a power calculation as is common in quantitative research; rather, sample sizes are guided by information power. We have amended the data analysis section of the paper to clarify accordingly: ‘The sample size was deemed large enough to provide adequate information power due to the narrow focus of the study, specific sample population, and clinical expertise of the research team’, referenced as Malterud K, Siersma VD, Guassora AD. Sample Size in Qualitative Interview Studies: Guided by Information Power. Qual Health Res. 2016 Nov;26(13):1753-1760. [lines 162-164]

The conclusion of this qualitative study has enabled the design of an interventional study in a new cohort, comparing a novel holistic screening tool with a subjective global assessment for all participants, as we herald in the final paragraph of the discussion. As would be expected, this follow-on study includes a full quantitative statistical analysis. We have made this more explicit: ‘The authors plan to develop a novel screening tool for ALD, that will be evaluated in comparison to a comprehensive subjective global assessment’. [lines 343-344].

3)    The concept of malnutrition relies in the amount of ingested food, its quality or the frequency of meals?

Malnutrition in ALD can occur because of both reduced meal frequency and quality; however, reduced meal quality due to anorexia, early satiety, ascites and gastrointestinal symptoms is more common. In this study, clinicians, patients, and carers were asked open questions about factors influencing nutrition, rather than specific questions regarding meal frequency or quality (Table 1 and 2). However, we did note that symptoms may impact the ability to shop and prepare meals, which could reflect meal frequency; however, measurement of meal frequency, for example using a validated food frequency questionnaire, was not the aim of the study. We also noted that finances may influence the amount of food that was affordable, which could influence food quality, although measurement of meal quality was also beyond the scope of the study. This is an interesting suggestion and we have added it to the discussion: ‘….and the factors that may differentially influence meal frequency or meal quality (may also benefit from future research)’. [line 335].

4) It seems that the main non-physiological factors relay in a depressing behavior and in the economical status. How these factors can be analyzed in depth?

The non-physiological factors that were identified in this study included financial status, level of food literacy, and level of sociological support. The aim of this study was not to establish a hierarchy of physiological or non-physiological factors. However, including these factors in a novel screening tool may allow a sensitivity analysis to determine the strength of the relationship between each factor and a subjective global assessment as the diagnostic evaluation of nutritional status. We have included in the last paragraph of the discussion: ‘The authors plan to develop a novel screening tool for ALD, that will be evaluated in comparison to a comprehensive subjective global assessment’. [lines 342-343].

Round 2

Reviewer 1 Report

Comments and Suggestions for Authors

Revised manuscript was well addressed to the reviewers' comments and well written.